# Is the lady's-slipper orchid (*Cypripedium calceolus*) likely to shortly become extinct in Europe?—Insights based on ecological niche modelling

**Marta Kolanowska**[1,2]*, **Anna Jakubska-Busse**[3]

**1** Department of Geobotany and Plant Ecology, Faculty of Biology and Environmental Protection, University of Lodz, Lodz, Poland, **2** Department of Biodiversity Research, Global Change Research Institute AS CR, Brno, Czech Republic, **3** Department of Botany, Institute of Environmental Biology, University of Wrocław, Wrocław, Poland

* martakolanowska@wp.pl

**Data Availability Statement:** All relevant data, including list of samples used in this study are provided as supplementary material.

## Abstract

Lady's-slipper orchid (*Cypripedium calceolus*) is considered an endangered species in most countries within its geographical range. The main reason for the decline in the number of populations of this species in Europe is habitat destruction. In this paper the ecological niche modelling approach was used to estimate the effect of future climate change on the area of niches suitable for *C. calceolus*. Predictions of the extent of the potential range of this species in 2070 were made using climate projections obtained from the Community Climate System Model for four representative concentration pathways: rcp2.6, rcp4.5, rcp6.0 and rcp8.5. According to these analyses all the scenarios of future climate change will result in the total area of niches suitable for *C. calceolus* decreasing. Considering areas characterized by a suitability of at least 0.4 the loss of habitat will vary between ca. 30% and 63%. The highest habitat loss of ca. 63% is predicted to occur in scenario rcp 8.5. Surprisingly, in the most damaging rcp 8.5 prediction the highest overlap between potential range of *C. calceolus* and its pollinators will be observed and in all other scenarios some pollinators will be available for this species in various geographical regions. Based on these results at least two approaches should be implemented to improve the chances of survival of *C. calceolus*. In view of the unavoidable loss of suitable habitats in numerous European regions, conservation activities should be intensified in areas where this species will still have suitable niches in the next 50 years. In addition, for *C. calceolus* ex-situ activities should be greatly increased so that it can be re-introduced in the remaining suitable areas.

## Introduction

Various statistical models are used to predict the spatial distribution of plant and animal species based on presence-only data [1–4]. This approach is also used in many conservation studies [5–6] such as evaluating the distributions or areas suitable for conservation [6–7] and

**Funding:** The research described here was supported by the Education Youth and Sports (MSMT) grant nr LO1415 (MK). https://app. dimensions.ai/details/grant/grant.6884710 The funder had no role in study design, data collection and analysis, decision to publish, or preparation of the manuscript.

**Competing interests:** The authors have declared that no competing interests exist.

identifying priority areas for conservation [8]. Unfortunately, species distribution models are rarely used for research on the largest angiosperm plant family, the Orchidaceae. Orchids are one of the most threatened groups as their complex life history make them particularly vulnerable to the effects of global environmental change [9–10].

*Cypripedium calceolus* is one of the most intensively studied European plants [11–20]. It is the only slipper orchid in Europe—just one additional species, *Cypripedium macranthos*, is found in Belarus. The geographical range of *C. calceolus* is relatively broad and includes Europe (except the extreme north and south), the Crimea, Mediterranean, Asia Minor, western and eastern Siberia, Far East of Russia and south of Sakhalin Island [21–23]. Lady's-slipper orchid used to be more widespread in Europe, but the number of its populations declined in the 19th century due to the over-collection for horticulture and habitat degradation [24].

Nowadays *C. calceolus* is considered as endangered in most countries within its range [21] is listed in Appendix II of the Convention on International Trade in Endangered Species of Wild Fauna and Flora (CITES) and also in Annex II of the Habitats Directive and under Appendix I of the Convention on the Conservation of European Wildlife and Natural Habitats (Bern Convention) [18] [23] [25].

Natural populations of this slipper orchid are included in Natura 2000 sites and other types of protected areas. This plant is also included on several national red lists and red data books as threatened [23]. In many countries this taxon is extremely rare, critically endangered and/or regionally extinct [19] [21] [23] and in others it is classified as Endangered (e.g. Croatia, Czech Republic, Hungary, Russia and Spain) or Vulnerable (e.g. Austria, Belarus, Denmark, France, Germany, Lithuania, Slovakia and Switzerland [23] [26]. Noteworthy, *C. calceolus* is a differential species for the unique Polish Kashubian region plant community—*Fagus sylvatica-Cypripedium calceolus* [27].

In the last century a significant decline in the number of populations of this species was recorded in almost all of Europe. This is due to many reasons, above all habitat destruction, especially expansion of agriculture, inappropriate forest management such as clearcutting, widespread use of herbicides and pesticides, equipment that can severely compact the soil, road and trail construction and collecting [23]. In addition, according to Rankou, & Bilz [23] browsing and grazing can pose a threat in two different ways: overgrazing affects individuals whereas the abandonment of traditional grazing leads to natural succession and therefore an increase in competition for this orchid. The replacement of natural forest with spruce plantations has caused habitat degradation as the soil is de-calcified and this species is linked to calcareous soils [23].

Climate change, especially the lack of rainfall and dry seasons, as well as the fires recorded in recent years in almost all regions of Europe may be responsible for the decline in the number of specimens in natural populations of *C. calceolus*. Currently, numerous (sub)populations of this species in various regions of Europe are fragmented remnants and genetically isolated. This raises the question—what is the future of this orchid? Is this species becoming extinct before our eyes? For instance, the dramatic decline of *C. calceolus* populations in Lower Silesia (SW Poland) was recorded and documented for over 100 years [28–30]. Among the 30 localities of this species, 12 were listed after 1945, in 2012 only 9 of them were confirmed [31], however, in 2019 only 7 were confirmed.

Furthermore, the occurrence of *C. calceolus* may be limited in the future by extinction or modification of the geographical ranges or ecology of its pollinators. While the lady's-slipper orchid is self-compatible, insects are required to transfer pollen to the stigma [32] as the position of the stigma and anthers prevent self-pollination [33]. Recent studies indicated that global warming can disturb the pollination of other European orchid, *Ophrys sphegodes*, which is pollinated by *Andrena nigroaenea* [34]. Noteworthy, *Andrena* bees are also one of the most

important pollinators of *C. calceolus*. Undoubtedly, reproduction success is crucial for the long-term existence of the surviving populations [35].

The aim of this study was to evaluate the predicted effect of global warming on the distribution and coverage of the ecological niches that are currently suitable for *C. calceolus* in Europe as well as to estimate the impact of the climate changes on the availability of its pollinators.

## Materials & methods

### List of localities

The database of *C. calceolus* localities was compiled based on information in public facilities (e.g. GBIF, ukrbin.com, tela-botanica.org, iNaturalist, redbook.minpriroda.gov.by, WildSlovenia, Portale della Flora d'Italia, naturamediterraneo.com), published articles and books [21] [24] [36–49], conservation reports (FAO, Berne Convention Resolution 6, Krajowy plan ochrony gatunku obuwik pospolity; [50]) and field observations made by Jakubska-Busse. While identification of numerous orchid species requires taxonomic skills and experience and for such taxa using information derived from public databases is not recommended, *C. calceolus* is the most spectacular terrestrial orchid in Europe which can be easily recognized even by amateur naturalists.

The list of *C. calceolus* pollinators was compiled based on available literature data [15] [18] [24] [51–55]. Information about distribution of 21 from a total of 24 reported insect species was gathered from GBIF. Due to the lack of sufficient, precise information about distribution of *Musca autumnalis*, *Andrena fulvicrus* and *Andrena ovina*, these species were not included in the analyses. Pollinators of *C. calceolus* belong mostly to Hymenoptera, but two, *Chrysotoxum festivum* and *Syrphus ribesii*, represent Syrphidae, Diptera. *Nomada panzeri* is classified within Nomadinae, and *Colletes cunicularius* within Colletidae. Seven species, *Halictus tumularum*, *Lasioglossum albipes*, *L. calceatum*, *L. fratellum*, *L. fulvicorne*, *L. morio*, and *L. quadrinotatum* belong to Halictidae. The highest number of pollinators represent genus *Andena* (Andrenidae)–*Andrena carantonica*, *A. cineraria*, *A. flavipes*, *A. fucata*, *A. haemorrhoa*, *A. helvola*, *A. nigroaenea*, *A. praecox*, *A. scotica*, *A. tibialis*.

### Ecological niche modelling

The ecological niche modelling was done using the maximum entropy method in MaxEnt version 3.3.2 [56–58] based on presence-only observations of this species. From the total of 932 locations of *C. calceolus* gathered during the study (Fig 1, S1 Table) the duplicate presence records (records within the same grid cell) were removed using MaxEnt. Considering pollinators input data, due to the various coordinate precision used in public databases only records georeferenced with the precision of at least 2 km were used to guarantee correct location of the observation in the grid cell. For data thinning and to minimize geographical overrepresentation of the samples, the initial catalogue was then reduced to include only records distanced one from another for at least 10 km and again the duplicate presence records (records within the same grid cell) were removed using MaxEnt. The final database included 519 localities of *Chrysotoxum festivum*, 2040 of *Syrphus ribesii*, 739 of *Nomada panzeri*, 621 of *Colletes cunicularius*, 1004 of *Halictus tumularum*, 1151 of *Lasioglossum albipes*, 1122 of *L. calceatum*, 940 of *L. fratellum*, 469 of *L. fulvicorne*, 699 of *L. morio*, 123 of *L. quadrinotatum*, 273 of *Andrena carantonica*, 1201 of *A. cineraria*, 325 of *A. flavipes*, 731 of *A. fucata*, 1477 of *A. haemorrhoa*, 1258 of *A. helvola*, 694 of *A. nigroaenea*, 486 of *A. praecox*, 421 of *A. scotica*, and 189 of *A. tibialis* (Figs 2–4; S2 Table).

For the modelling bioclimatic variables in 2.5 arc-minutes ($\pm$ 21.62 km$^2$ at the equator) of interpolated climate surface were used. This approach was justified considering the precision

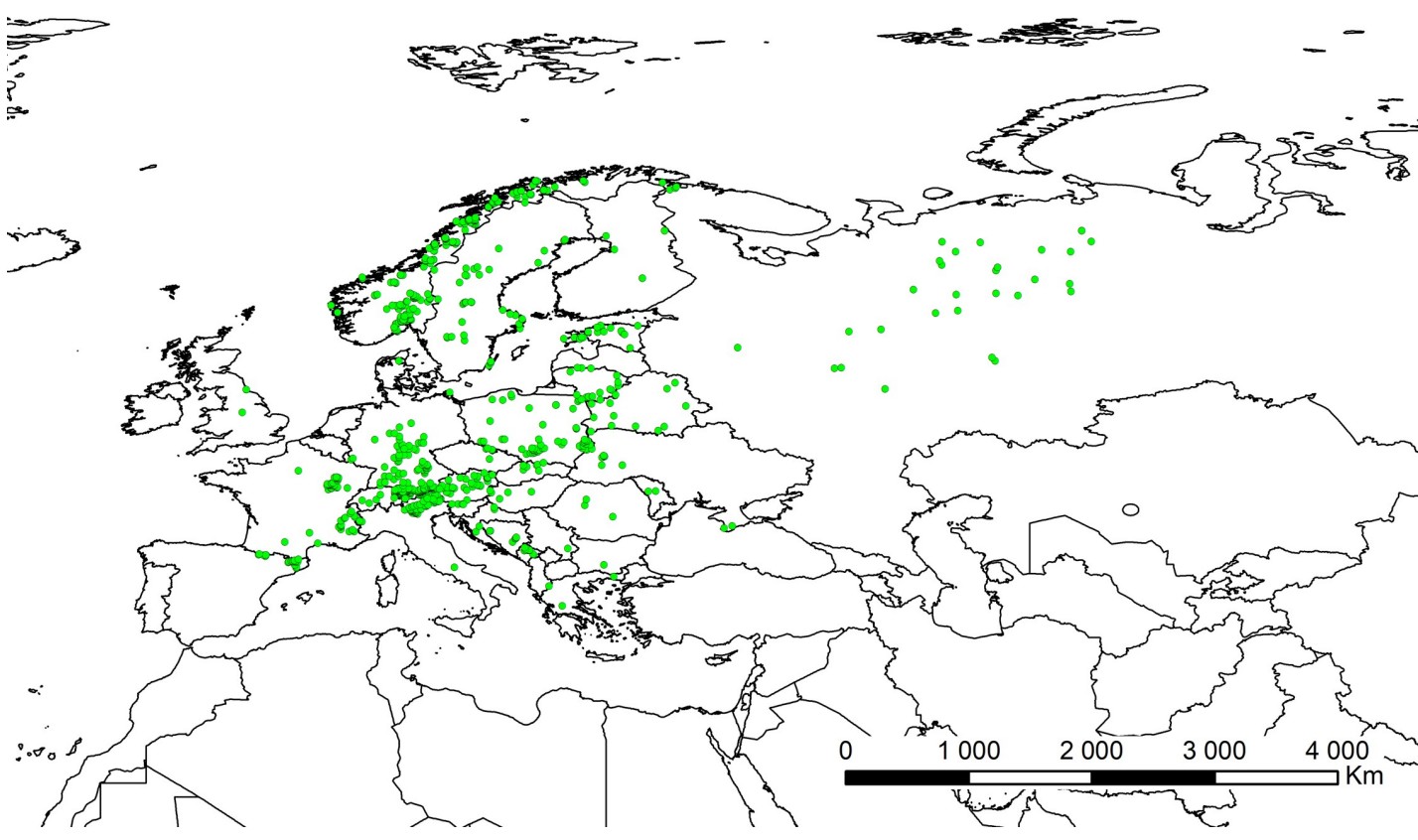

**Fig 1. Localities of *C. calceolus* georeferenced in this study.** Map was generated in ArcGis 9.3 (ESRI, 2006).

of georeferenced records of both, orchid and its pollinators–the information available in public databases was not sufficient to conduct further analyses using more detailed maps (in 30 arc-seconds). Because some previous studies [59] indicated that usage of a restricted area in ENM analysis is more reliable than calculating habitat suitability on the global scale, the area of the analysis was restricted to 78.83˚N-34.08˚N– 13.12˚W-77.29˚E.

In this study the most widespread source of data for ecological studies was used. WorldClim [60] is commonly applied to produce species distribution models (> 15000 citations). Of 19 climatic variables ("bioclims", Table 1) available in WorldClim (version 1.4, www.worldclim. org) seven were removed as they were significantly correlated with one another (above 0.9) as evaluated by Pearsons' correlation coefficient computed using ENMTools v1.3 [61]. As a result of the reduction of multi-collinearity the following variables were excluded from further analyses: bio6, bio7, bio9, bio10, bio11, bio16 and bio17. Because MaxEnt is relatively robust against collinear variables [62] we decided not to remove other data from the analyses. The most recent research results suggested that the strategy of excluding highly correlated variables has little influence on models derived from MaxEnt [63].

Predictions of the future extent of the climatic niches of *C. calceolus* in 2070 were made using climate projections obtained from the Community Climate System Model (CCSM4) which was commonly used in previous studies on orchids (e.g. [64–65]). Four representative concentration pathways (RCPs: rcp2.6, rcp4.5, rcp6.0, rcp8.5) were analyzed. These pathways are trajectories adopted by the Intergovernmental Panel on Climate Change (IPCC) for its fifth Assessment Report in 2014. These four scenarios describe potential future climate of the world assuming various amounts of greenhouse gases will be emitted. The RCPs are named

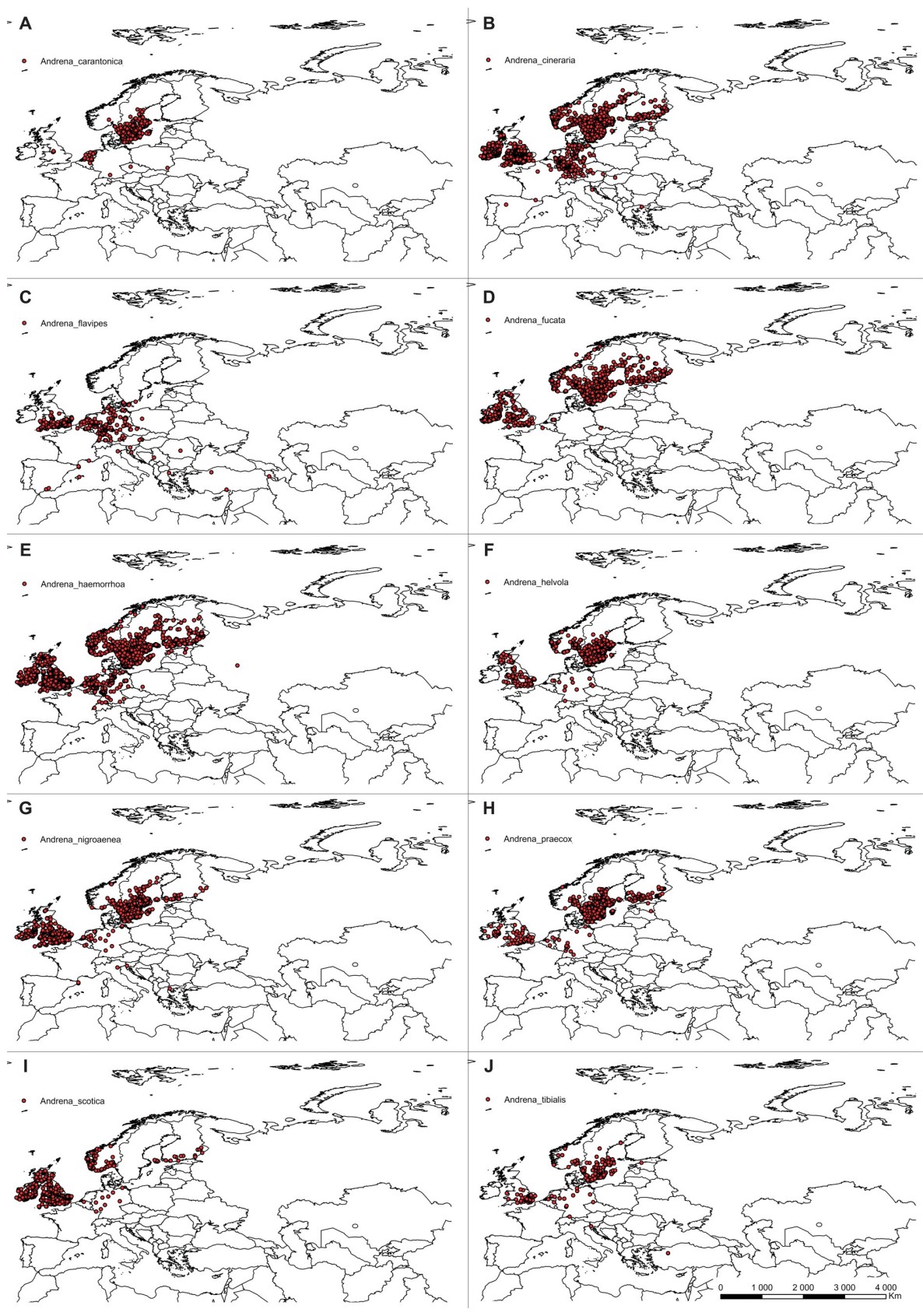

**Fig 2. Localities of pollinators georeferenced in this study.** *Andrena carantonica* (**A**), *Andrena cineraria* (**B**), *Andrena flavipes* (**C**), *Andrena fucata* (**D**), *Andrena haemorrhoa* (**E**), *Andrena helvola* (**F**), *Andrena nigroaenea* (**G**), *Andrena praecox* (**H**), *Andrena scotica* (**I**), and *Andrena tibialis* (**J**) georeferenced in this study. Map was generated in ArcGis 9.3 (ESRI, 2006).

after a possible range of radiative forcing values in 2100, relative to pre-industrial values (+2.6, +4.5, +6.0 and +8.5 W/m2 respectively; [66–67]). These climate projections were used in several previous studies on threatened plants (e.g. [68–69]) and endangered animals (e.g. [70–71]).

In all analyses the maximum number of iterations was set to 10000 and convergence threshold to 0.00001. The neutral (= 1) regularization multipler value and auto features were used.

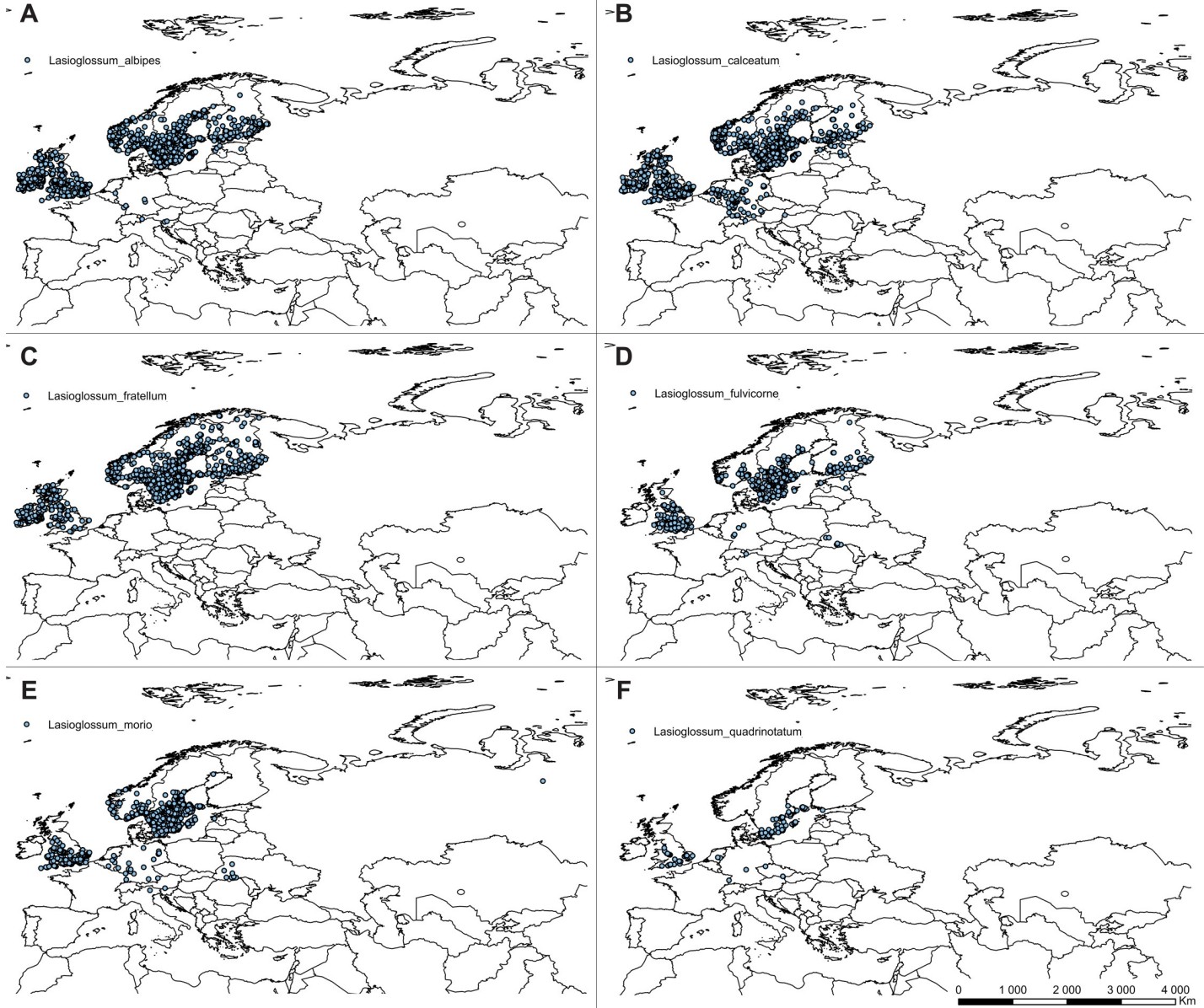

**Fig 3. Localities of pollinators georeferenced in this study.** *Lasioglossum albipes* (**A**), *Lasioglossum calceatum* (**B**), *Lasioglossum fratellum* (**C**), *Lasioglossum fulvicorne* (**D**), *Lasioglossum morio* (**E**), *Lasioglossum quadrinotatum* (**F**) georeferenced in this study. Map was generated in ArcGis 9.3 (ESRI, 2006).

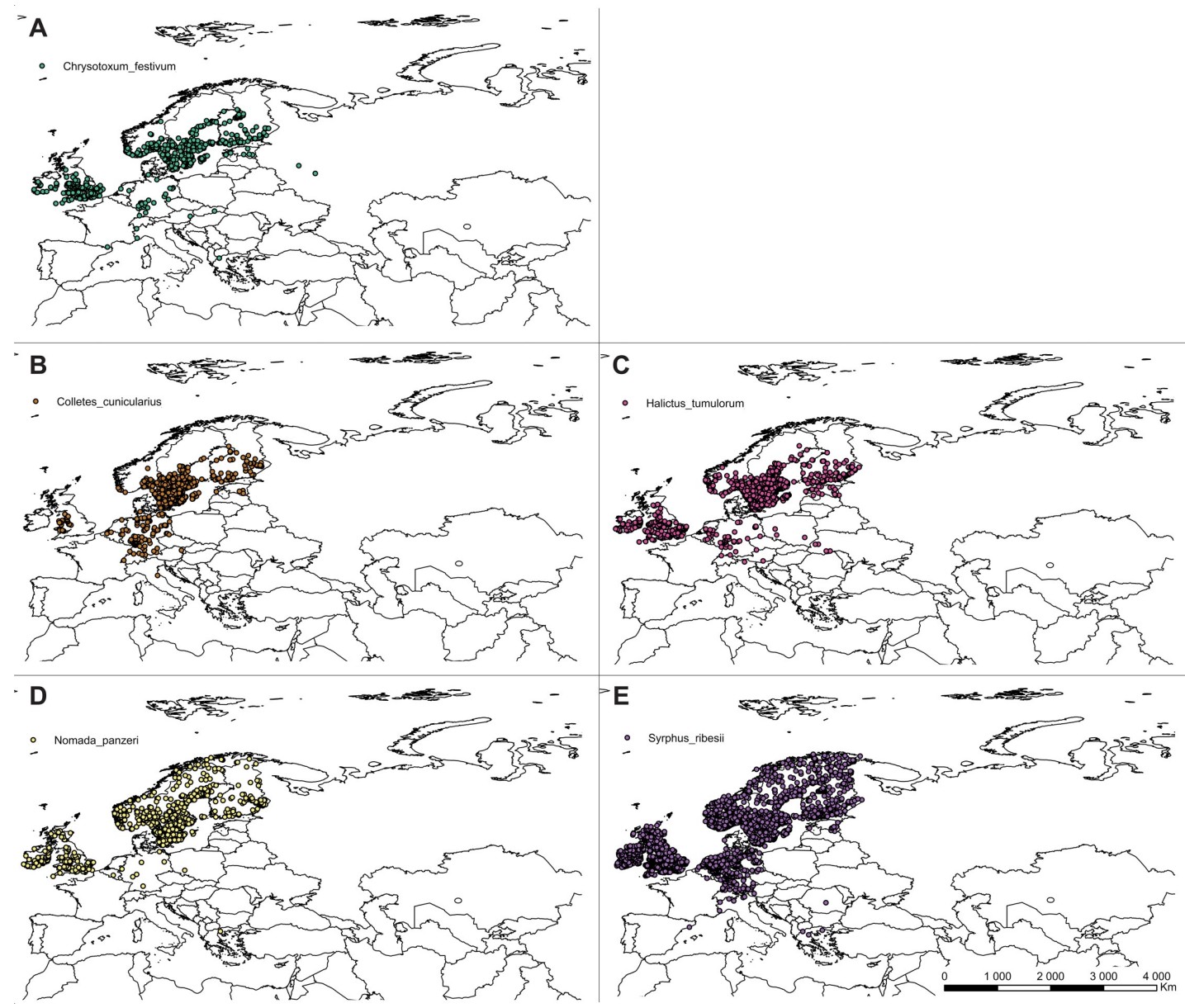

**Fig 4. Localities of pollinators georeferenced in this study.** *Chrysotoxum festivum* (**A**), *Colletes cunicularius* (**B**), *Halictus tumulorum* (**C**), *Nomada panzeri* (**D**), *Syrphus ribesii* (**E**) georeferenced in this study. Map was generated in ArcGis 9.3 (ESRI, 2006).

All samples were added to the backgroud. The "random seed" option which provided a random test partition and background subset for each run was applied. 10% of the samples were used as test points. While often larger test samples are used in species distribution models [72], we followed Oraie et al. [73], Ashraf et al. [74], and Tobeña et al. [75] in our analyses. The run was performed as a bootstrap with 100 replicates, and the output was set to logistic. While bootstrap is also recommended for small-sample analyses we followed Slater & Michael [76] in our modelling. All operations on GIS data were carried out on ArcGis 10.6 (Esri, Redlands, CA, USA). The evaluation of the created models was made using the most common metric—the area under the curve (AUC; [77–79].

**Table 1. Codes of climatic variables developed by Hijmans et al. [60].**

| Code | Description |
|------|-------------|
| bio1 | Annual Mean Temperature |
| bio2 | Mean Diurnal Range = Mean of monthly (max temp − min temp) |
| bio3 | Isothermality (bio2/bio7) * 100 |
| bio4 | Temperature Seasonality (standard deviation * 100) |
| bio5 | Max Temperature of Warmest Month |
| bio6 | Min Temperature of Coldest Month |
| bio7 | Temperature Annual Range (bio5—bio6) |
| bio8 | Mean Temperature of Wettest Quarter |
| bio9 | Mean Temperature of Driest Quarter |
| bio10 | Mean Temperature of Warmest Quarter |
| bio11 | Mean Temperature of Coldest Quarter |
| bio12 | Annual Precipitation |
| bio13 | Precipitation of Wettest Month |
| bio14 | Precipitation of Driest Month |
| bio15 | Precipitation Seasonality (Coefficient of Variation) |
| bio16 | Precipitation of Wettest Quarter |
| bio17 | Precipitation of Driest Quarter |
| bio18 | Precipitation of Warmest Quarter |
| bio19 | Precipitation of Coldest Quarter |

To visualize the climatic preferences of *C. calceolus* the predicted niche occupancy profiles (PNOs) were created using the Phyloclim package [80]. SDMtoolbox 2.3 for ArcGIS [81–82] was used to visualize changes in the distribution of suitable niches of studied orchid and its pollinators caused by the global warming [81]. To compare distribution model created for current climatic conditions with future models all SDMs were converted into binary rasters and projected using Albers EAC (as implemented in SDMtoolbox 2.3) as projection. The presence threshold was estimated individually for each species based on the values of grids in which studied species occur in models created using present-time. For *C. calceolus*, *Chrysotoxum festivum*, *Syrphus ribesii*, *Nomada panzeri*, *Colletes cunicularius*, *Halictus tumularum*, *Lasioglossum albipes*, *L. calceatum*, *L. fratellum*, *L. fulvicorne*, *L. morio*, *Andrena carantonica*, *A. cineraria*, *A. fucata*, *A. haemorrhoa*, *A. helvola*, *A. nigroaenea*, *A. praecox*, and *A. scotica* the threshold for presence was set as 0.4. The habitat suitability of at least 0.3 was considered as sufficient for the occurrence of *Andrena flavipes*, *A. tibialis, and Lasioglossum quadrinotatum*. Furthermore, to estimate the pollinator availability, the binary models of predicted range of *C. calceolus* were compared with future distribution of its pollinators to calculate the number of grid cells in which both orchid and insect could occur.

## Results

### Predicted distribution of *C. calceolus*

The average training AUC for the replicate runs received scores of 0.912–0.914, which indicates that the MaxEnt models are very reliable (Table 2).

Created map of the potential distribution of *C. calceolus* (Fig 5) is consistent with the known location of populations of this species. However, some additional regions (e.g. Eastern Carpathians and western valley of the Danube river) were indicated by the ENM analysis as potentially suitable for this orchid.

**Table 2. The average training AUC for the replicate runs for created models of *C. calceolus*.**

| Scenario | AUC score |
|---|---|
| present | 0.914 |
| rcp2.6 | 0.913 |
| rcp4.5 | 0.914 |
| rcp6.0 | 0.912 |
| rcp8.5 | 0.914 |

The total area of niches suitable for *C. calceolus* will decrease in 2070 according to three of four scenarios of future climate change analyzed (Fig 6). Considering areas characterized by a suitability of at least 0.4 the loss of habitat will vary between ca. 30% and 63%. Surprisingly scenario rcp 6.0 will be slightly less harmful than rcp 4.5. The highest habitat loss of ca. 63% is predicted in rcp 8.5. In this scenario relatively large suitable areas will still be available in Scandinavia but the niche coverage in the Pyrenees and the Alps will be significantly smaller than currently. *C. calceolus* will almost disappear from the Carpathians and there will be no suitable niches for this orchid in the Apennines, Balkans, lowlands of Baltic countries and valleys of the major European rivers. The changes in the distribution of the coverage of suitable niches of *C. calceolus* are presented in Fig 7 and Table 3.

Of the bioclimatic factors analyzed the most important variables influencing the distribution of *C. calceolus* are temperature seasonality (bio4) and precipitation in the warmest quarter (bio18; Table 4). Somewhat less significant for the occurrence of this species is precipitation in the driest month (bio14). The PNO profiles of *C. calceolus* for these three vital variables are presented in Fig 8.

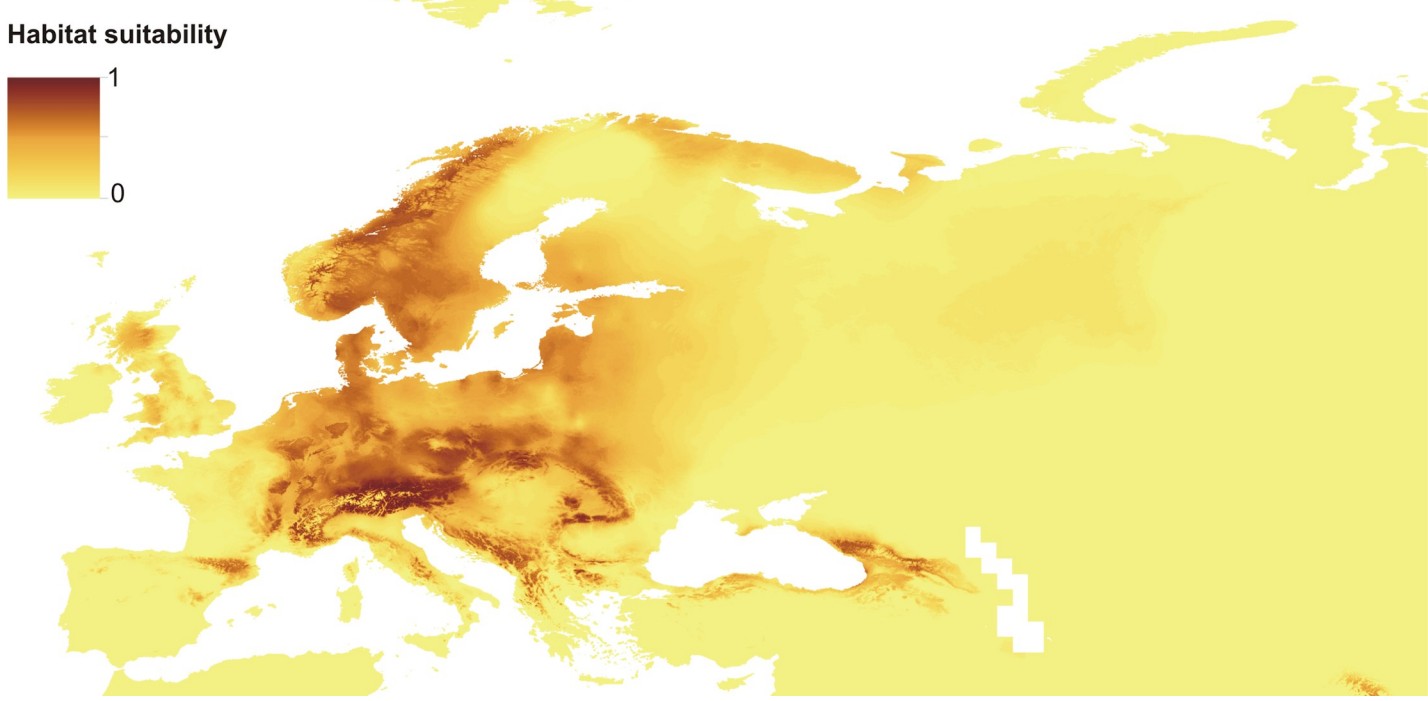

**Fig 5. Present distribution of suitable niches of *C. calceolus*.** Map was generated in ArcGis 9.3 (ESRI, 2006).

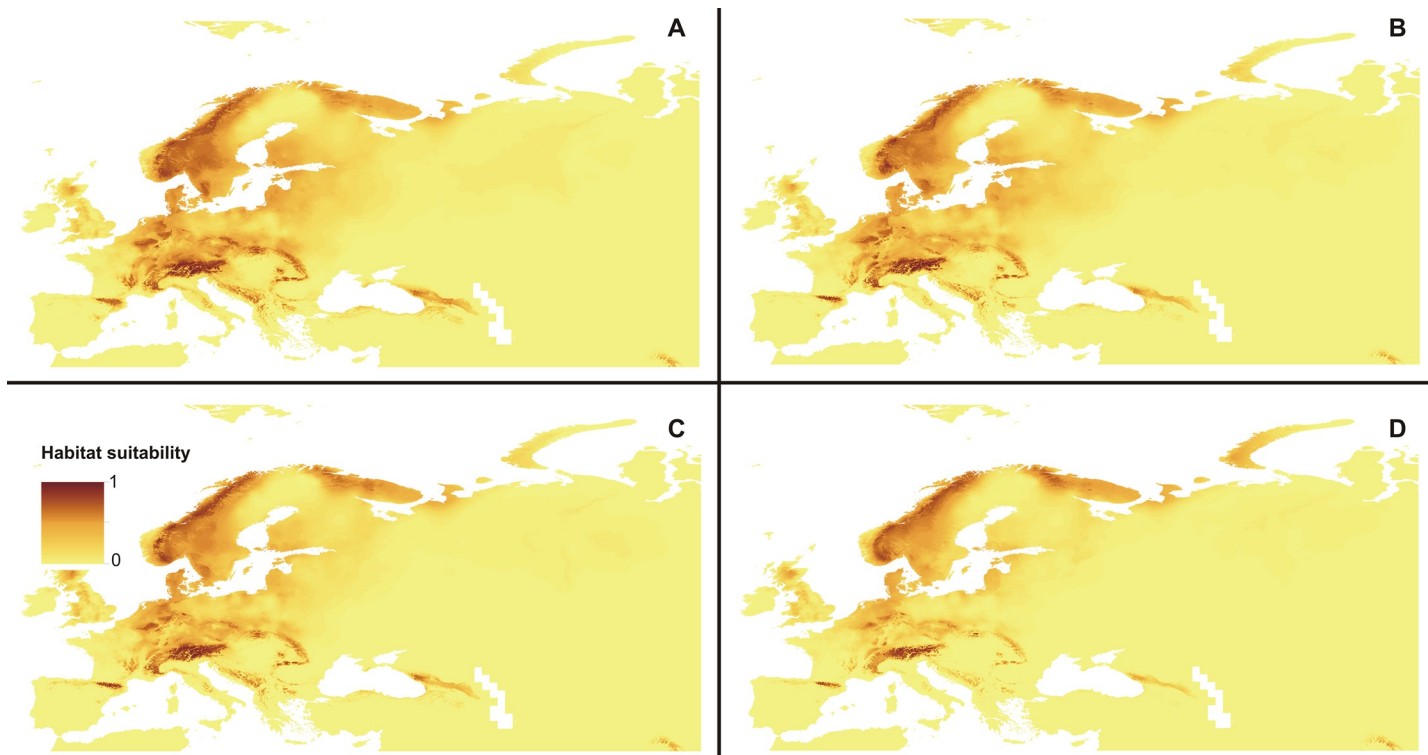

**Fig 6. Future distribution of suitable niches of *C. calceolus*.** Estimations based on rcp2.6 scenario (**A**), rcp4.5 scenario (**B**), rcp6.0 scenario (**C**) and rcp8.5 scenario (**D**). Maps were generated in ArcGis 9.3 (ESRI 2006).

### Predicted availability of *C. calceolus* pollinators

The average training AUC for the replicate runs received scores of 0.878–0.989, which indicates that the MaxEnt models are very reliable (Table 5). The predicted potential ranges of all studied insect species are presented as S1–S4 Figs.

Except of a single case of *Andrena helvola* (only rcp4.5 scenario) all pollinators of *C. calceolus* will face habitat loss caused by the climate changes (Table 6). The highest decrease of 410327– 786796 km$^2$ in the coverage of the suitable niches will be observed in *Syrphus ribesii*. Generally, the rcp8.5 scenario will cause the most significant damages in the available habitats of studied species. In this scenario Diptera representatives, *Chrysotoxum festivum Syrphus ribesii* will lose respectively 78995 km$^2$ and 786796 km$^2$ of their current niche coverage. The potential range of the only Nomadinae species, *Nomada panzeri*, will be smaller for 479387 km$^2$. The decrease of 387973 km$^2$ will be observed in *Colletes cunicularius*. Within Halictidae the most significant habitat loss is predicted for *Lasioglossum calceatum* (474419 km$^2$) and within *Andrena* representatives the highest range contraction will be observed in *Andrena cineraria* (624529 km$^2$).

Considering the predicted range overlap of *C. calceolus* and studied insects (Fig 9, Table 7), the highest pollination potential in the future will be attributed to *Syrphus ribesii* which will occur in 45.85–66.81% of *C. calceolus* range. The global warming will almost exclude the possibility of pollen transfer by *Lasioglossum quadrinotatum* which will overlap with the lady's-slipper orchid in just 0.27–2.59% of the orchid niches coverage. Similar situation will be observed in *Andrena carantonica* (0.42–7.31%) and *A. scotica* (2.06–7.86%). Surprisingly, the highest overlap between *C. calcolus* and its pollinators is expected in rcp8.5 scenario. In this generally unsuitable climatic conditions seven of the studied insects will be available for the lady's-slipper orchid in more than 40% of its range—*Andrena cineraria, A. fucata, A. haemorrhoa,*

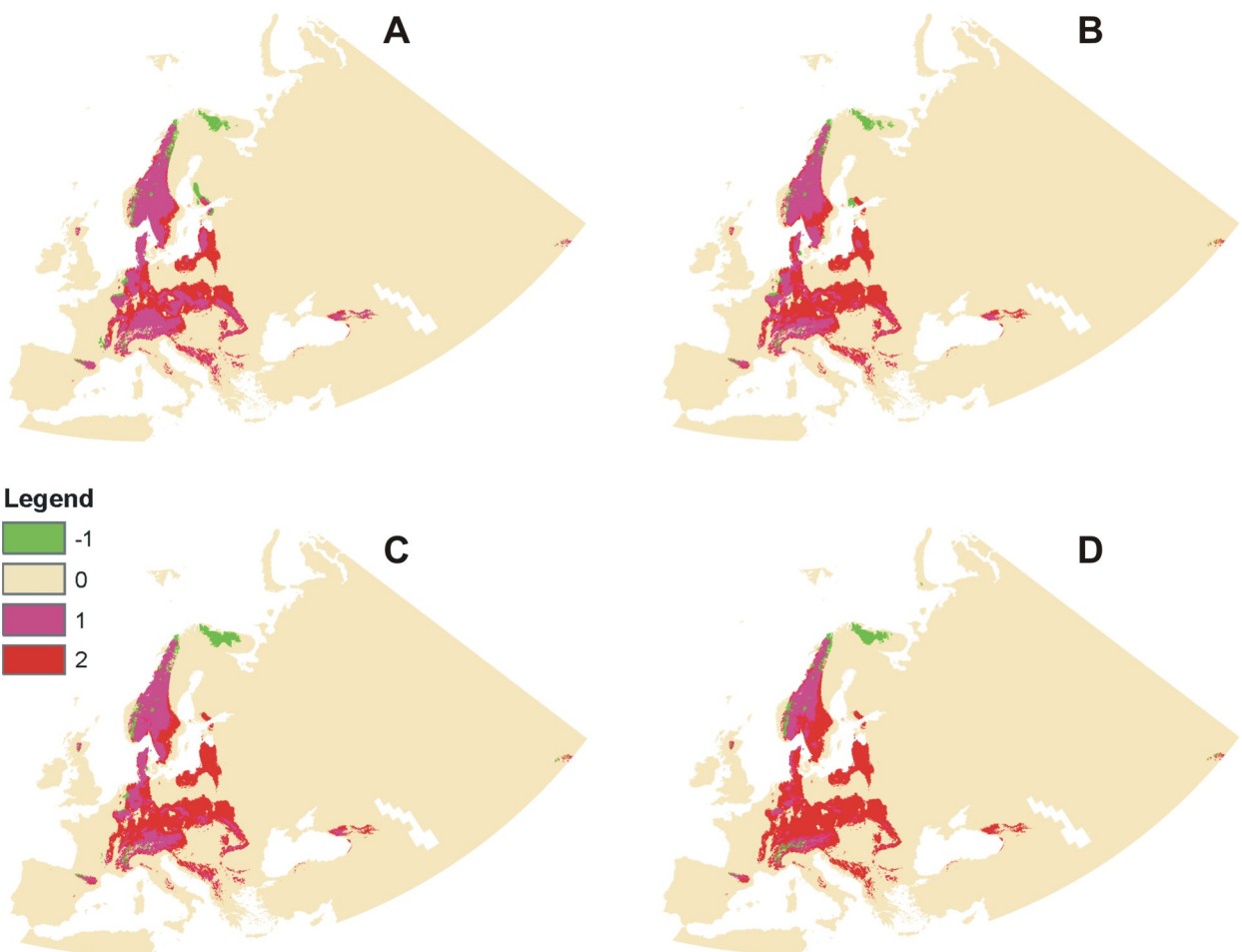

**Fig 7. Changes in the distribution of the coverage of suitable niches of *C. calceolus* in various climate change scenarios.** Rcp2.6 (**A**), rcp4.5 (**B**), rcp6.0 (**C**) and rcp8.5 (**D**). -1 = range expansion, 0 = no occupancy (absence in both), 1 = no change (presence in both), 2 = range contraction. Maps were generated in ArcGis 10.6 (ESRI). Albers EAC projection.

*Lasioglossum albipes*, *L. fratellum*, *Nomada panzeri*, and *Syrphus ribesii*. *Nomada panzeri*, the only representative of Nomadinae, will occur in 25.30–52.96% of the predicted range of *C. calceolus*, while *Colletes cunicularius* (Colletidae) will be able to pollinate orchid populations in 10.10–21.46% of the range. *Lasioglossum fratellum* will be the most important pollinator of *C. calceolus* within Halictidae–this species will be available in 22.74–51.64% of the orchid range. Considering *Andrena* species, the most significant contribution to the orchid propagation will be attributed to *Andrena fucata* which can occur in 18.36–44.60% of the lady's-slipper orchid range.

**Table 3. Changes in the coverage of suitable niches of *C. calceolus*.**

| Scenario | Number of grid cells ≥ 0.4 | Range expansion [km²] | Range contraction [km²] |
|---|---|---|---|
| present | 125382 | - | - |
| rcp2.6 | 88063 | 135057.6446 | 690919.2031 |
| rcp4.5 | 67337 | 107792.1356 | 933432.8332 |
| rcp6.0 | 70680 | 136271.5200 | 939595.5852 |
| rcp8.5 | 46517 | 131135.8933 | 1236528.1833 |

**Table 4. The estimates of relative contributions of the environmental variables to the Maxent models.**

| scenario | variable 1 | variable 2 | variable 3 |
|---|---|---|---|
| present | bio4 (40.8) | bio18 (30.9) | bio14 (15.1) |
| rcp2.6 | bio4 (43.5) | bio18 (33.3) | bio14 (12.8) |
| rcp4.5 | bio4 (43.4) | bio18 (31.1) | bio14 (14.8) |
| rcp6.0 | bio4 (43.5) | bio18 (31.9) | bio14 (11.9) |
| rcp8.5 | bio4 (44.6) | bio18 (30.4) | bio14 (13.5) |

## Discussion

Based on the results of this research the area of niches suitable for *C. calceolus* will significantly decrease under all the currently available scenarios of climate change. The ENM was used previously in a very few studies on the effect of climate change on orchids. While global warming is predicted to negatively affect European species of *Dactylorhiza* [64] suitable niches for holomycoheterotrophic *Neottia nidus-avis* and *Epipogium aphyllum* are predicted to become more widespread [83]. In the recent regional study Kaye et al. [84] evaluated the probability of extinction of American *Cypripedium fasciculatum* based on population size, time between

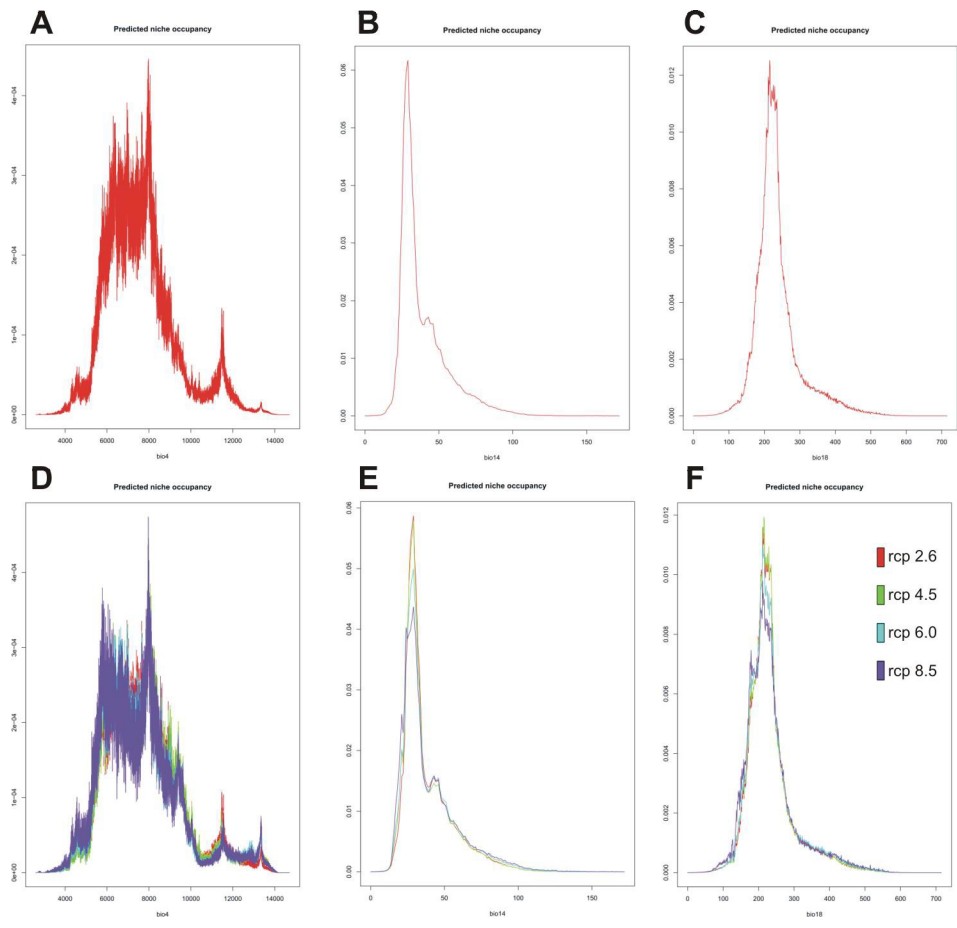

**Fig 8.** Predicted niche occupancy profiles created for present models (A-C), and future climate change scenarios (D-F). Diagrams generated in RStudio using the Phyloclim package (Heibl & Calenge, 2013). Albers EAC projection.

**Table 5. The average training AUC for the replicate runs for created models of *C. calceolus* pollinators [SD–standard deviation].**

| Species | Scenario | | | | |
|---|---|---|---|---|---|
| | present | rcp2.6 | rcp4.5 | rcp6.0 | rcp8.5 |
| *Andrena carantonica* | 0.979, SD = 0.001 | 0.980, SD = 0.001 | 0.980, SD = 0.001 | 0.980, SD = 0.001 | 0.980, SD = 0.001 |
| *Andrena cineraria* | 0.921, SD = 0.001 | 0.922, SD = 0.001 | 0.922, SD = 0.001 | 0.921, SD = 0.001 | 0.920, SD = 0.001 |
| *Andrena flavipes* | 0.969, SD = 0.001 | 0.969, SD = 0.001 | 0.969, SD = 0.002 | 0.968, SD = 0.001 | 0.969, SD = 0.001 |
| *Andrena fucata* | 0.947, SD = 0.001 | 0.946, SD = 0.001 | 0.947, SD = 0.001 | 0.948, SD = 0.001 | 0.947, SD = 0.001 |
| *Andrena haemorrhoa* | 0.914, SD = 0.001 | 0.914, SD = 0.001 | 0.912, SD = 0.001 | 0.914, SD = 0.001 | 0.916, SD = 0.001 |
| *Andrena helvola* | 0.955, SD = 0.001 | 0.955, SD = 0.001 | 0.950, SD = 0.001 | 0.956, SD = 0.001 | 0.955, SD = 0.001 |
| *Andrena nigroaenea* | 0.956, SD = 0.001 | 0.956, SD = 0.001 | 0.956, SD = 0.001 | 0.955, SD = 0.001 | 0.956, SD = 0.001 |
| *Andrena praecox* | 0.963, SD = 0.001 | 0.963, SD = 0.001 | 0.962, SD = 0.001 | 0.963, SD = 0.001 | 0.961, SD = 0.001 |
| *Andrena scotica* | 0.968, SD = 0.001 | 0.969, SD = 0.001 | 0.970, SD = 0.001 | 0.969, SD = 0.001 | 0.970, SD = 0.001 |
| *Andrena tibialis* | 0.979, SD = 0.002 | 0.978, SD = 0.002 | 0.976, SD = 0.002 | 0.979, SD = 0.002 | 0.979, SD = 0.002 |
| *Chrysotoxum festivum* | 0.955, SD = 0.001 | 0.953, SD = 0.001 | 0.956, SD = 0.001 | 0.955, SD = 0.002 | 0.956, SD = 0.001 |
| *Colletes cunicularius* | 0.954, SD = 0.001 | 0.954, SD = 0.001 | 0.955, SD = 0.001 | 0.953, SD = 0.001 | 0.955, SD = 0.001 |
| *Halictus tumulorum* | 0.935, SD = 0.001 | 0.935, SD = 0.001 | 0.935, SD = 0.001 | 0.937, SD = 0.001 | 0.936, SD = 0.001 |
| *Lasioglossum albipes* | 0.931, SD = 0.001 | 0.932, SD = 0.001 | 0.930, SD = 0.001 | 0.930, SD = 0.001 | 0.930, SD = 0.001 |
| *Lasioglossum calceatum* | 0.925, SD = 0.001 | 0.926, SD = 0.001 | 0.928, SD = 0.001 | 0.927, SD = 0.001 | 0.926, SD = 0.001 |
| *Lasioglossum fratellum* | 0.933, SD = 0.001 | 0.934, SD = 0.001 | 0.935, SD = 0.001 | 0.934, SD = 0.001 | 0.936, SD = 0.001 |
| *Lasioglossum fulvicorne* | 0.960, SD = 0.001 | 0.961, SD = 0.001 | 0.960, SD = 0.001 | 0.960, SD = 0.001 | 0.960, SD = 0.001 |
| *Lasioglossum morio* | 0.955, SD = 0.001 | 0.954, SD = 0.001 | 0.955, SD = 0.001 | 0.954, SD = 0.001 | 0.955, SD = 0.001 |
| *Lasioglossum quadrinotatum* | 0.988, SD = 0.001 | 0.988, SD = 0.001 | 0.989, SD = 0.001 | 0.989, SD = 0.001 | 0.988, SD = 0.001 |
| *Nomada panzeri* | 0.935, SD = 0.002 | 0.937, SD = 0.002 | 0.938, SD = 0.001 | 0.939, SD = 0.001 | 0.937, SD = 0.002 |
| *Syrphus ribesii* | 0.878, SD = 0.001 | 0.878, SD = 0.001 | 0.879, SD = 0.001 | 0.881, SD = 0.001 | 0.879, SD = 0.001 |

**Table 6. Loss of suitable niches [km$^2$] of studied pollinators of *C. calceolus* in various climate change scenarios.**

| Species | rcp2.6 | rcp4.5 | rcp6.0 | rcp8.5 |
|---|---|---|---|---|
| *Andrena carantonica*–total habitat loss | 106690.3 | 183650 | 183463.3 | 162565.9 |
| *Andrena cineraria*–total habitat loss | 270283.4 | 523591.1 | 458826.2 | 624529.6 |
| *Andrena flavipes*–total habitat loss | 23885.33 | 39871.14 | 130351.5 | 350735.3 |
| *Andrena fucata*–total habitat loss | 196535.8 | 287893.9 | 294411.5 | 301601.3 |
| *Andrena haemorrhoa*–total habitat loss | 315719.7 | 514010.9 | 470610.2 | 545384.9 |
| *Andrena helvola*–total habitat loss | 92721.41 | -226677 | 198814.1 | 123030.9 |
| *Andrena nigroaenea*–total habitat loss | 130706.4 | 289014.4 | 256669.3 | 416247.2 |
| *Andrena praecox*–total habitat loss | 26163.68 | 194388.1 | 193622.5 | 314188.3 |
| *Andrena scotica*–total habitat loss | 34175.26 | 83589.33 | 77314.53 | 141892.7 |
| *Andrena tibialis*–total habitat loss | 42579.01 | 68966.8 | 89845.45 | 134123.9 |
| *Chrysotoxum festivum*–total habitat loss | 8833.278 | 51094.82 | 56753.34 | 78995.28 |
| *Colletes cunicularius*–total habitat loss | 155955 | 379625.5 | 345879.8 | 387973.3 |
| *Halictus tumulorum*–total habitat loss | 101311.9 | 184285 | 154442.3 | 219468.7 |
| *Lasioglossum albipes*–total habitat loss | 237676.8 | 302703.2 | 320911.3 | 351575.7 |
| *Lasioglossum calceatum*–total habitat loss | 226695.9 | 369765.1 | 396395.7 | 474419.9 |
| *Lasioglossum fratellum*–total habitat loss | 248825.8 | 373257.3 | 350660.6 | 407059.1 |
| *Lasioglossum fulvicorne*–total habitat loss | 46295.34 | 167047.9 | 176497.5 | 234800.9 |
| *Lasioglossum morio*–total habitat loss | 107698.8 | 166357 | 172407.7 | 227162.8 |
| *Lasioglossum quadrinotatum*–total habitat loss | 105812.6 | 123890 | 105289.7 | 118567.6 |
| *Nomada panzeri*–total habitat loss | 246640.8 | 376450.8 | 357140.8 | 479387.4 |
| *Syrphus ribesii*–total habitat loss | 410327.2 | 657883.1 | 710023.7 | 786796.7 |

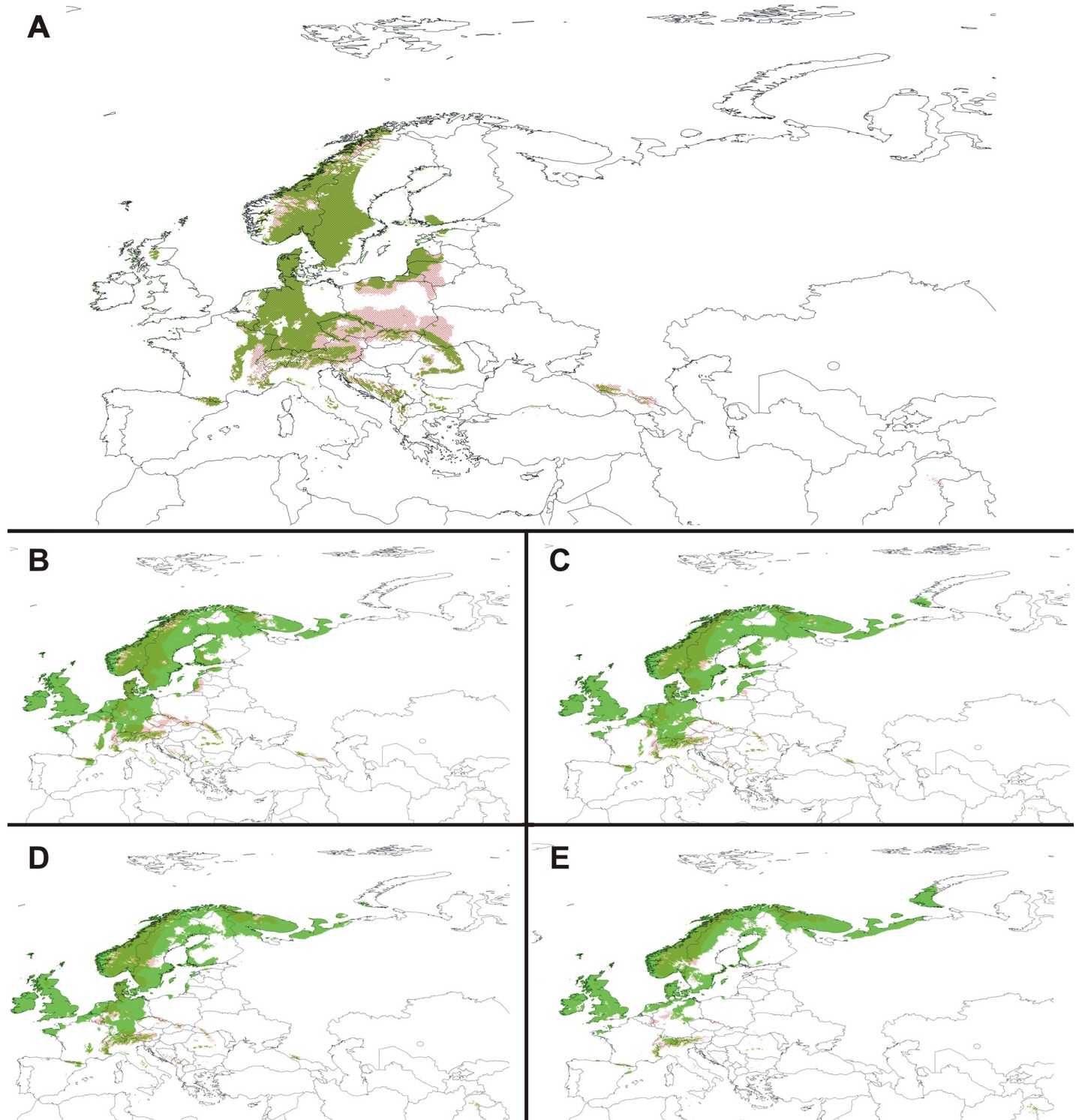

**Fig 9. Predicted niche overlap between *C. calceolus* (red diagonal shade) and its pollinators (green shade) in various climate change scenarios.** Present time (**A**), rcp2.6 scenario (**B**), rcp4.5 scenario (**C**), rcp6.0 scenario (**D**) and rcp8.5 scenario (**E**). Maps were generated in ArcGis 10.6 (ESRI).

**Table 7. The number of grid cells where both *C. calceolus* and specific pollinator can occur in various climate change scenarios.**

| | present | | rcp26 | | rcp45 | | rcp60 | | rcp85 | |
|---|---|---|---|---|---|---|---|---|---|---|
| | number of common grid cells | part of *C. calceolus* range | number of common grid cells | part of *C. calceolus* range | number of common grid cells | part of *C. calceolus* range | number of common grid cells | part of *C. calceolus* range | number of common grid cells | part of *C. calceolus* range |
| *Andrena carantonica* | 9163 | 0.073 | 4132 | 0.047 | 281 | 0.004 | 467 | 0.007 | 628 | 0.014 |
| *Andrena cineraria* | 40077 | 0.320 | 27903 | 0.317 | 14583 | 0.217 | 19745 | 0.279 | 20514 | 0.441 |
| *Andrena flavipes* | 20249 | 0.161 | 11444 | 0.130 | 9442 | 0.140 | 6442 | 0.091 | 1229 | 0.026 |
| *Andrena fucata* | 23024 | 0.184 | 21360 | 0.243 | 14047 | 0.209 | 17928 | 0.254 | 20748 | 0.446 |
| *Andrena haemorrhoa* | 32401 | 0.258 | 23890 | 0.271 | 13568 | 0.201 | 18670 | 0.264 | 18969 | 0.408 |
| *Andrena helvola* | 15286 | 0.122 | 10581 | 0.120 | 20474 | 0.304 | 5472 | 0.077 | 7354 | 0.158 |
| *Andrena nigroaenea* | 15130 | 0.121 | 10503 | 0.119 | 3167 | 0.047 | 6195 | 0.088 | 3073 | 0.066 |
| *Andrena praecox* | 19039 | 0.152 | 19895 | 0.226 | 9360 | 0.139 | 11974 | 0.169 | 7646 | 0.164 |
| *Andrena scotica* | 2586 | 0.021 | 2680 | 0.030 | 1206 | 0.018 | 2646 | 0.037 | 3657 | 0.079 |
| *Andrena tibialis* | 16841 | 0.134 | 13401 | 0.152 | 8424 | 0.125 | 8745 | 0.124 | 7687 | 0.165 |
| *Chrysotoxum festivum* | 19656 | 0.157 | 18478 | 0.210 | 11385 | 0.169 | 13988 | 0.198 | 16490 | 0.354 |
| *Colletes cunicularius* | 26466 | 0.211 | 18899 | 0.215 | 6803 | 0.101 | 9200 | 0.130 | 8711 | 0.187 |
| *Halictus tumulorum* | 22195 | 0.177 | 19054 | 0.216 | 9620 | 0.143 | 13448 | 0.190 | 13636 | 0.293 |
| *Lasioglossum albipes* | 23108 | 0.184 | 20308 | 0.231 | 14743 | 0.219 | 19327 | 0.273 | 20080 | 0.432 |
| *Lasioglossum calceatum* | 28482 | 0.227 | 23254 | 0.264 | 14971 | 0.222 | 17002 | 0.241 | 18402 | 0.396 |
| *Lasioglossum fratellum* | 28508 | 0.227 | 26083 | 0.296 | 18417 | 0.274 | 21548 | 0.305 | 24021 | 0.516 |
| *Lasioglossum fulvicorne* | 18992 | 0.151 | 20538 | 0.233 | 10093 | 0.150 | 10924 | 0.155 | 8576 | 0.184 |
| *Lasioglossum morio* | 17897 | 0.143 | 12864 | 0.146 | 6254 | 0.093 | 7365 | 0.104 | 6848 | 0.147 |
| *Lasioglossum quadrinotatum* | 2013 | 0.016 | 241 | 0.003 | 711 | 0.011 | 1084 | 0.015 | 1203 | 0.026 |
| *Nomada panzeri* | 31722 | 0.253 | 27968 | 0.318 | 21341 | 0.317 | 23132 | 0.327 | 24635 | 0.530 |
| *Syrphus ribesii* | 58772 | 0.469 | 41718 | 0.474 | 30877 | 0.459 | 34297 | 0.485 | 31078 | 0.668 |

surveys, and elevation. This research revealed that 39–52% of *C. fasciculatum* populations are likely extinct. In our study we investigated exclusively the impact of climate changes on European *C. calceolus* but the similar loss of habitats was predicted (30–63%).

As evaluated in this study temperature seasonality and precipitation in the warmest quarter are crucial climatic factors limiting the distribution of *C. calceolus*. Therefore, it is not surprising that global warming will cause a decrease in the availability of suitable niches for this species in Europe. According to the National Aeronautics and Space Administration (NASA) rising temperatures will intensify the world's water cycle and increase evaporation. As a result, storm-affected regions will experience increases in precipitation, while the increased risk of drought is predicted for areas located far away from storm tracks. As a result of global warming the heat capacity of the surface layer will increase due to loss of sea ice. Dwyer, Biasutti & Sobel [85] reported that when seasonality of surface temperature is considered, the phase delay and amplitude decrease are strongest at high latitudes and will drive the global response.

Noteworthy, while our analyses included the evaluation of possible effects of the global warming on the distribution of suitable niches of both, studied orchid and its pollinators, there are other factors that can increase the extinction rate of the lady's-slipper orchid. Like all other Orchidaceae representatives, *C. calceolus* requires mycorrhizal fungi for germination and seedling nutrition. Distribution of *Cypripedium* can be hereby limited by mycorrhizal specificity [86] and while this relationship is not primarily limited by fungal distribution but by genetically controlled specialization [87], the further studies could be improved and include also analysis of the possible changes of European mycobiota.

Based on our results at least two approaches should be implemented to improve the chances of survival of *C. calceolus*. In view of the unavoidable loss of suitable habitats in numerous European regions, conservation activities over the next 50 years should be concentrated in areas where there are still suitable niches for this species. The prioritization of preservation zones is suggested by Seaton et al. [88] but their proposal is mainly for the most biodiverse, tropical regions of the world.

In addition, for *C. calceolus* ex-situ activities should be carried out at a large scale. Seed storage will enable the cultivation of this rare orchid in the future and successful reintroduction into the wild. This method is already effectively being used to re-establish the lady's slipper orchid in Britain [89] and could be used in the future to introduce *C. calceolus* in the remaining suitable areas.

## Conclusions

Our research results indicated significant loss (30%-63%) of suitable habitat of *C. calceolus* in 2070, but the pollinator availability should not further limit the chance of survival of this species. The highest decrease of niches coverage was predicted in rcp 8.5 scenario of future climate change. Temperature seasonality and precipitation in the warmest quarter are crucial climatic factors limiting the distribution of *C. calceolus*, therefore, it is not surprising that global warming will cause a decrease in the availability of suitable niches for the studied species in Europe. Based on our results at least two approaches should be implemented to improve the chances of survival of the lady's-slipper orchid in Europe. In view of the unavoidable loss of suitable habitats in numerous European regions, conservation activities over the next 50 years should be concentrated in areas where there are still suitable niches for this species. Furthermore, for *C. calceolus* ex-situ activities, e.g. steed storage, should be carried out at a large scale. Noteworthy, while both orchid and its pollinators were included in our analyses, the extinction of the lady's-slipper orchid may be further driven by the modification of local mycobiota.

## Supporting information

**S1 Table. Localities of *C. calceolus* gathered in this study.**
(XLS)

**S2 Table. Localities of pollinators of *C. calceolus* gathered in this study.**
(XLSX)

**S1 Fig. Changes in the distribution of the coverage of suitable niches of *C. calceolus* pollinators in 2070 based on rcp2.6 climate change scenario.** *Andrena carantonica* (**A**), *Andrena cineraria* (**B**), *Andrena flavipes* (**C**), *Andrena fucata* (**D**), *Andrena haemorrhoa* (**E**), *Andrena helvola* (**F**), *Andrena nigroaenea* (**G**), *Andrena praecox* (**H**), *Andrena scotica* (**I**), *Andrena tibialis* (**J**), *Chrysotoxum festivum* (**K**), *Colletes cunicularius* (**L**), *Halictus tumulorum* (**M**), *Lasioglossum albipes* (**N**), *Lasioglossum calceatum* (**O**), *Lasioglossum fratellum* (**Q**), *Lasioglossum fulvicorne* (**P**), *Lasioglossum morio* (**R**), *Lasioglossum quadrinotatum* (**S**), *Nomada panzeri* (**T**),

*Syrphus ribesii* (**U**). -1 = range expansion, 0 = no occupancy (absence in both), 1 = no change (presence in both), 2 = range contraction.
(TIF)

**S2 Fig. Changes in the distribution of the coverage of suitable niches of *C. calceolus* pollinators in 2070 based on rcp4.5 climate change scenario.** *Andrena carantonica* (**A**), *Andrena cineraria* (**B**), *Andrena flavipes* (**C**), *Andrena fucata* (**D**), *Andrena haemorrhoa* (**E**), *Andrena helvola* (**F**), *Andrena nigroaenea* (**G**), *Andrena praecox* (**H**), *Andrena scotica* (**I**), *Andrena tibialis* (**J**), *Chrysotoxum festivum* (**K**), *Colletes cunicularius* (**L**), *Halictus tumulorum* (**M**), *Lasioglossum albipes* (**N**), *Lasioglossum calceatum* (**O**), *Lasioglossum fratellum* (**Q**), *Lasioglossum fulvicorne* (**P**), *Lasioglossum morio* (**R**), *Lasioglossum quadrinotatum* (**S**), *Nomada panzeri* (**T**), *Syrphus ribesii* (**U**). -1 = range expansion, 0 = no occupancy (absence in both), 1 = no change (presence in both), 2 = range contraction.
(TIF)

**S3 Fig. Changes in the distribution of the coverage of suitable niches of *C. calceolus* pollinators in 2070 based on rcp6.0 climate change scenario.** *Andrena carantonica* (**A**), *Andrena cineraria* (**B**), *Andrena flavipes* (**C**), *Andrena fucata* (**D**), *Andrena haemorrhoa* (**E**), *Andrena helvola* (**F**), *Andrena nigroaenea* (**G**), *Andrena praecox* (**H**), *Andrena scotica* (**I**), *Andrena tibialis* (**J**), *Chrysotoxum festivum* (**K**), *Colletes cunicularius* (**L**), *Halictus tumulorum* (**M**), *Lasioglossum albipes* (**N**), *Lasioglossum calceatum* (**O**), *Lasioglossum fratellum* (**Q**), *Lasioglossum fulvicorne* (**P**), *Lasioglossum morio* (**R**), *Lasioglossum quadrinotatum* (**S**), *Nomada panzeri* (**T**), *Syrphus ribesii* (**U**). -1 = range expansion, 0 = no occupancy (absence in both), 1 = no change (presence in both), 2 = range contraction.
(TIF)

**S4 Fig. Changes in the distribution of the coverage of suitable niches of *C. calceolus* pollinators in 2070 based on rcp8.5 climate change scenario.** *Andrena carantonica* (**A**), *Andrena cineraria* (**B**), *Andrena flavipes* (**C**), *Andrena fucata* (**D**), *Andrena haemorrhoa* (**E**), *Andrena helvola* (**F**), *Andrena nigroaenea* (**G**), *Andrena praecox* (**H**), *Andrena scotica* (**I**), *Andrena tibialis* (**J**), *Chrysotoxum festivum* (**K**), *Colletes cunicularius* (**L**), *Halictus tumulorum* (**M**), *Lasioglossum albipes* (**N**), *Lasioglossum calceatum* (**O**), *Lasioglossum fratellum* (**Q**), *Lasioglossum fulvicorne* (**P**), *Lasioglossum morio* (**R**), *Lasioglossum quadrinotatum* (**S**), *Nomada panzeri* (**T**), *Syrphus ribesii* (**U**). -1 = range expansion, 0 = no occupancy (absence in both), 1 = no change (presence in both), 2 = range contraction.
(TIF)

## Acknowledgments

We would like to thank prof. Maxim A. Dzhus (Belarusian State University), Peter Efimov, PhD (Komarov Botanical Institute of the Russia Academy of Sciences), Myroslav Shevera, PhD (National Academy of Sciences of Ukraine), Mindaugas Lapele, PhD (Dzukija National Park, Lithuania) and Spyros Tsiftsis, PhD (Eastern Macedonia and Thrace Institute of Technology) for providing data on the distribution of *C. calceolus*. We are grateful to Prof. Anthony F.G. Dixon (University of East Anglia) for the valuable comments on the manuscript. Thanks are also due to Mr Jacek Stefaniak, MSc (University of Wrocław) for helpful discussions and to Giuseppe Brundu for valuable comments on the manuscript.

## Author Contributions

**Conceptualization:** Marta Kolanowska, Anna Jakubska-Busse.

**Data curation:** Marta Kolanowska, Anna Jakubska-Busse.

**Formal analysis:** Marta Kolanowska.

**Funding acquisition:** Marta Kolanowska.

**Investigation:** Marta Kolanowska, Anna Jakubska-Busse.

**Methodology:** Marta Kolanowska, Anna Jakubska-Busse.

**Project administration:** Marta Kolanowska.

**Resources:** Marta Kolanowska, Anna Jakubska-Busse.

**Software:** Marta Kolanowska.

**Supervision:** Marta Kolanowska.

**Validation:** Marta Kolanowska, Anna Jakubska-Busse.

**Visualization:** Marta Kolanowska.

**Writing – original draft:** Marta Kolanowska, Anna Jakubska-Busse.

**Writing – review & editing:** Marta Kolanowska, Anna Jakubska-Busse.

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
