## [Decision Letter · Decision Letter 0]

10 Dec 2019

PONE-D-19-27369

Is the lady's-slipper orchid (Cypripedium calceolus) likely to shortly become extinct in Europe? - insights based on ecological niche modelling

PLOS ONE

Dear Dr. Kolanowska,

Thank you for submitting your manuscript to PLOS ONE. After careful consideration, we feel that it has merit but does not fully meet PLOS ONE’s publication criteria as it currently stands. Therefore, we invite you to submit a revised version of the manuscript that addresses the points raised during the review process.

We would appreciate receiving your revised manuscript by Jan 24 2020 11:59PM. To enhance the reproducibility of your results, we recommend that if applicable you deposit your laboratory protocols in protocols.io, where a protocol can be assigned its own identifier (DOI) such that it can be cited independently in the future. For instructions see: http://journals.plos.org/plosone/s/submission-guidelines#loc-laboratory-protocols

We look forward to receiving your revised manuscript.

Kind regards,

Jana Müllerová, Ph.D

Academic Editor

PLOS ONE

Journal Requirements:

https://www.iucnredlist.org/species/162021/43316125

In your revision ensure you cite all your sources (including your own works), and quote or rephrase any duplicated text outside the methods section. Further consideration is dependent on these concerns being addressed.

3. Thank you for including your competing interests statement; "no"

Additional Editor Comments (if provided):

Dear Author, your manuscript is interesting and worth publishing but needs more work, make sure in your re-submission to adequately address all issues raised by the reviewer, especially trying to enlarge the data collection following the suggestions.

Reviewers' comments:

Reviewer's Responses to Questions

**Comments to the Author**

1. Is the manuscript technically sound, and do the data support the conclusions?

Reviewer #1: Partly

2. Has the statistical analysis been performed appropriately and rigorously? 

Reviewer #1: Yes

3. Have the authors made all data underlying the findings in their manuscript fully available?

Reviewer #1: Yes

4. Is the manuscript presented in an intelligible fashion and written in standard English?

Reviewer #1: Yes

5. Review Comments to the Author

Reviewer #1: General comments

An interesting Ms that, in my opinion, is it worth to be published after major revision.

I would strongly suggest to address more carefully the distribution of Cypripedium calceolus and all the references and data-bases that could be used. I had the impression the total the number of presence records could be increased after a more careful verification of all the available sources (for a number of Countries). A number of additional potential databases and refences is provided as an example. I think it is worth to improve the data collection and re-run the model.

I think it is very important to start with the better available distribution dataset, if not the level of uncertainly in the model rises too much. In fact, there are already a number of uncertainties areas due to, e.g.: (1) using a single model and not, for example, ensemble modelling techniques; (2) excluding from the model the future changes in land use and the future distribution of beech forest. A number of studies consider the Fagus sylvatica s.l. as a species sensitive to climatic extremes, especially drought and water deficit, which reduces its competitive advantage over less drought-sensitive species, and this will ultimately result in forest vegetation transformation, (3) including in the model the areas that are not and will not be suitable in the future (this could be achieved with masking le land use such as the urban areas that are not of interest).

In the discussion session I would suggest to discuss the results of the present research also in comparison with the modelling of Cypripedium fasciculatum in US done by ThomasN.Kaye (Population extinctions driven by climate change, population size, and time since observation may make rare species databases inaccurate, - PLoSONE 14(10):e0210378.https://doi.org/10.1371/journal.pone.0210378).

Minor comments

LL 63-64: “Appendix I of the Convention on the Conservation of European Wildlife and 64 Natural Habitats of Bern Convention” – please rephrase: Convention on the Conservation of European Wildlife and Natural Habitats (Bern Convention);

L 67: “forms” – types;

L 161: “Figure 1. Localities of C. calceolus georeferenced in this study” -. This map seems not to take into account the distribution of the species in Italy, in particular in the NW, probably due to the fact that Italy is not enough included in GBIF. I would suggest to have a look at: http://dryades.units.it/floritaly/index.php?procedure=taxon_page&tipo=all&id=8099 and http://www.naturachevale.it/wp-content/uploads/2016/06/Cypripedium-calceolus-L_new.pdf

http://fll-italia.it/UploadDocs/6103_G_Perazza___M__Decarli_Perazza_p_129.pdf

http://www.storianaturale.org/anp/PDF%20ANP/24_2003_Isaja%20Dotti_Le%20Orchidee%20spontanee%20della%20Val%20di%20Susa%20Primi%20dati%20sulla%20distribuzione%20di%20tre.pdf

https://www.naturamediterraneo.com/forum/topic.asp?TOPIC_ID=118207

http://www.fondazionemcr.it/UploadDocs/15_art08.pdf

http://www.isprambiente.gov.it/public_files/direttiva-habitat/Manuale-140-2016.pdf (page 128)

I also would strongly suggest to take into consideration the distribution and the reference in the EuroMed PlantBase at:

http://ww2.bgbm.org/euroPlusMed/PTaxonDetailOccurrence.asp?NameId=48350&PTRefFk=8000000

In addition, detailed distribution records are included in the “Action plan for Cypripedium Calceolus in Europe (Nature and Environment No. 100) (1999), Council of Europe” and in TIIU KULL, Journal of Ecology, 1999, 87, 913-924 (BIOLOGICAL FLORA OF THE BRITISH ISLES) and: http://powo.science.kew.org/taxon/urn:lsid:ipni.org:names:320700-2#distribution-map

- LAZARE, J.-J., J. MIRAIXES & L. VILLAR (1987). Cypripedium calceolus (Orchidaceae) en el Pirineo. Anales Jard. Bol. Madrid43(2): 375-382.;

http://citeseerx.ist.psu.edu/viewdoc/download?doi=10.1.1.602.3796&rep=rep1&type=pdf

https://www.conservacionvegetal.org/wp-content/uploads/publicaciones/Catalogo%20de%20especies%20amenazadas%20en%20Aragon.pdf

L 368: “suitable habitat” – The modelling do not consider habitats but climate, so I would not use the word “habitat” here; in fact in L 333 the term “niches suitable” is used.

6. PLOS authors have the option to publish the peer review history of their article (what does this mean?). If published, this will include your full peer review and any attached files.

Reviewer #1: Yes: Giuseppe Brundu

---

## [Author Response · Author response to Decision Letter 0]

16 Dec 2019

Additional Editor Comments (if provided):

Dear Author, your manuscript is interesting and worth publishing but needs more work, make sure in your re-submission to adequately address all issues raised by the reviewer, especially trying to enlarge the data collection following the suggestions.

Reviewers' comments:

Reviewer's Responses to Questions

Comments to the Author

1. Is the manuscript technically sound, and do the data support the conclusions?

Reviewer #1: Partly

2. Has the statistical analysis been performed appropriately and rigorously?

Reviewer #1: Yes

3. Have the authors made all data underlying the findings in their manuscript fully available?

Reviewer #1: Yes

4. Is the manuscript presented in an intelligible fashion and written in standard English?

Reviewer #1: Yes

5. Review Comments to the Author

Reviewer #1: General comments

An interesting Ms that, in my opinion, is it worth to be published after major revision.

I would strongly suggest to address more carefully the distribution of Cypripedium calceolus and all the references and data-bases that could be used. I had the impression the total the number of presence records could be increased after a more careful verification of all the available sources (for a number of Countries). A number of additional potential databases and refences is provided as an example. I think it is worth to improve the data collection and re-run the model.

I think it is very important to start with the better available distribution dataset, if not the level of uncertainly in the model rises too much. In fact, there are already a number of uncertainties areas due to, e.g.: (1) using a single model and not, for example, ensemble modelling techniques; (2) excluding from the model the future changes in land use and the future distribution of beech forest. A number of studies consider the Fagus sylvatica s.l. as a species sensitive to climatic extremes, especially drought and water deficit, which reduces its competitive advantage over less drought-sensitive species, and this will ultimately result in forest vegetation transformation, (3) including in the model the areas that are not and will not be suitable in the future (this could be achieved with masking le land use such as the urban areas that are not of interest).

In the discussion session I would suggest to discuss the results of the present research also in comparison with the modelling of Cypripedium fasciculatum in US done by ThomasN.Kaye (Population extinctions driven by climate change, population size, and time since observation may make rare species databases inaccurate, - PLoSONE 14(10):e0210378.https://doi.org/10.1371/journal.pone.0210378).

Authors: Corrected.

Minor comments

LL 63-64: “Appendix I of the Convention on the Conservation of European Wildlife and 64 Natural Habitats of Bern Convention” – please rephrase: Convention on the Conservation of European Wildlife and Natural Habitats (Bern Convention);

Authors: Corrected.

L 67: “forms” – types;

Authors: Corrected.

L 161: “Figure 1. Localities of C. calceolus georeferenced in this study” -. This map seems not to take into account the distribution of the species in Italy, in particular in the NW, probably due to the fact that Italy is not enough included in GBIF. I would suggest to have a look at: http://dryades.units.it/floritaly/index.php?procedure=taxon_page&tipo=all&id=8099 and http://www.naturachevale.it/wp-content/uploads/2016/06/Cypripedium-calceolus-L_new.pdf

http://fll-italia.it/UploadDocs/6103_G_Perazza___M__Decarli_Perazza_p_129.pdf

http://www.storianaturale.org/anp/PDF%20ANP/24_2003_Isaja%20Dotti_Le%20Orchidee%20spontanee%20della%20Val%20di%20Susa%20Primi%20dati%20sulla%20distribuzione%20di%20tre.pdf

https://www.naturamediterraneo.com/forum/topic.asp?TOPIC_ID=118207

http://www.fondazionemcr.it/UploadDocs/15_art08.pdf

http://www.isprambiente.gov.it/public_files/direttiva-habitat/Manuale-140-2016.pdf (page 128)

I also would strongly suggest to take into consideration the distribution and the reference in the EuroMed PlantBase at:

http://ww2.bgbm.org/euroPlusMed/PTaxonDetailOccurrence.asp?NameId=48350&PTRefFk=8000000

In addition, detailed distribution records are included in the “Action plan for Cypripedium Calceolus in Europe (Nature and Environment No. 100) (1999), Council of Europe” and in TIIU KULL, Journal of Ecology, 1999, 87, 913-924 (BIOLOGICAL FLORA OF THE BRITISH ISLES) and: http://powo.science.kew.org/taxon/urn:lsid:ipni.org:names:320700-2#distribution-map

- LAZARE, J.-J., J. MIRAIXES & L. VILLAR (1987). Cypripedium calceolus (Orchidaceae) en el Pirineo. Anales Jard. Bol. Madrid43(2): 375-382.;

http://citeseerx.ist.psu.edu/viewdoc/download?doi=10.1.1.602.3796&rep=rep1&type=pdf

https://www.conservacionvegetal.org/wp-content/uploads/publicaciones/Catalogo%20de%20especies%20amenazadas%20en%20Aragon.pdf

Authors: We have already used several sources provided by the Reviewer and these were cited in the previous version of ms (e.g. Lazare et al. 1986, Kull 1999, Devilliers-Terschuren 1999 – Action plan, etc.). We could not use all data suggested by the Reviewer due to the lack of possibility of precise georeferencing. Not all links provided by the Reviewer included specific locations but just general information about occurrence of C. calceolus (e.g. http://powo.science.kew.org/taxon/urn:lsid:ipni.org:names:320700-2#distribution-map). However, we included additional 43 Italian records based on:

- Perezza G., Decarli Perezza M. 2002. Cartogrfia orchidee trdentine (COT): Cypripedium calceolus L. e Liparis loeselii (L.) Rich., specie citate nella directiva habitat della CEE.

- https://www.naturamediterraneo.com

- Perazza G. 1995. Cartografia della orchidee (Orchidaceae) spontanee in Trentino-Alto Adige (Italia). Ricerca Sull'erbario dell'Universita di firence (FI). Ann. Mus. Civ. Rovereto 11: 231-256. 

- Pedrini P., Brambilla M., Bertolli A., Prosse F. 2014. Definizione di "linee guida provinciali" per l’attuazione dei monitoraggi nei siti trentinidella rete Natura 2000. LIFE+T.E.N - Azione A5 

- Isaja A., Dotti L. 2003. Le orchidee spontanee della Val di Susa (Piemonte-Italia) primi dati sulla distribuzione di tre orchidee rare: Cypripedium Calceolus L. (1735), Corallorhiza Trifida Chatelain (1760) e Aceras anthropophorum R.Br ex Aiton fil. (1814). Riv. Piem. St. Nat. 24: 205-215.

All models for C. calceolus were run again and all statistics were calculated using new outcomes.

L 368: “suitable habitat” – The modelling do not consider habitats but climate, so I would not use the word “habitat” here; in fact in L 333 the term “niches suitable” is used.

Authors: Corrected.

6. PLOS authors have the option to publish the peer review history of their article (what does this mean?). If published, this will include your full peer review and any attached files.

Do you want your identity to be public for this peer review? For information about this choice, including consent withdrawal, please see our Privacy Policy.

Reviewer #1: Yes: Giuseppe Brundu

---

## [Decision Letter · Decision Letter 1]

15 Jan 2020

Is the lady's-slipper orchid (Cypripedium calceolus) likely to shortly become extinct in Europe? - insights based on ecological niche modelling

PONE-D-19-27369R1

Dear Dr. Kolanowska,

We are pleased to inform you that your manuscript has been judged scientifically suitable for publication and will be formally accepted for publication once it complies with all outstanding technical requirements.

With kind regards,

Jana Müllerová, Ph.D

Academic Editor

PLOS ONE

Additional Editor Comments (optional):

All issues of previous version were addressed and by my opinion, the manuscript is now ready for publication. I have no further comments.

Reviewers' comments:

Reviewer's Responses to Questions

**Comments to the Author**

1. If the authors have adequately addressed your comments raised in a previous round of review and you feel that this manuscript is now acceptable for publication, you may indicate that here to bypass the “Comments to the Author” section, enter your conflict of interest statement in the “Confidential to Editor” section, and submit your "Accept" recommendation.

Reviewer #1: All comments have been addressed

2. Is the manuscript technically sound, and do the data support the conclusions?

Reviewer #1: Yes

3. Has the statistical analysis been performed appropriately and rigorously? 

Reviewer #1: Yes

4. Have the authors made all data underlying the findings in their manuscript fully available?

Reviewer #1: Yes

5. Is the manuscript presented in an intelligible fashion and written in standard English?

Reviewer #1: Yes

6. Review Comments to the Author

Reviewer #1: I am not native English speaker, so I do not feel enough qualified to evaluate point 5 above. However, I had no problems in reading and understanding the Ms and I have not found any major mistake.

7. PLOS authors have the option to publish the peer review history of their article (what does this mean?). If published, this will include your full peer review and any attached files.

Reviewer #1: Yes: Giuseppe Brundu

---

## [Editor Report · Acceptance letter]

21 Jan 2020

PONE-D-19-27369R1 

Is the lady's-slipper orchid (*Cypripedium calceolus*) likely to shortly become extinct in Europe? - insights based on ecological niche modelling 

Dear Dr. Kolanowska:

I am pleased to inform you that your manuscript has been deemed suitable for publication in PLOS ONE. Congratulations! Your manuscript is now with our production department. 

With kind regards,

on behalf of

Dr. Jana Müllerová 

Academic Editor

PLOS ONE